# Verifiable Natural Language to Linear Temporal Logic Translation: A Benchmark Dataset and Evaluation Suite

## Abstract

Empirical evaluation of state-of-the-art natural language (NL) to temporal logic (TL) translation systems reveals near-perfect performance on existing benchmarks. However, current studies only measure the accuracy of the *translation* of NL logic into formal TL, ignoring a system's capacity to *ground* atomic propositions into new scenarios or environments. This is a critical feature, necessary for the *verification* of resulting formulas in a concrete state space. In this paper, we introduce the **Verifiable Linear Temporal Logic Benchmark (VLTL-Bench)**, a unifying benchmark for automated NL-to-LTL translation. The dataset consists of three unique state spaces and thousands of diverse natural language specifications and their corresponding formal temporal logic specifications. Moreover, the benchmark contains sample traces to verify the temporal logic expressions. While the benchmark directly supports end-to-end evaluation, we observe that many frameworks decompose the process into i) lifting, ii) grounding, iii) translation, and iv) verification. The benchmark provides ground truths after each of these steps to enable researchers to improve and evaluate different substeps of the overall problem. Using the benchmark, we evaluate several state-of-the-art NL-to-TL translation models and frameworks, including nl2spec, NL2TL, NL2LTL, Lang2LTL, sequence-to-sequence translation, and various LLM prompting techniques. Our evaluation confirms that existing work is capable of reliably performing lifting and translation with high accuracy, while it exposes their struggles to ground the translation into a state space, which stems from the lack of existing datasets.

## 1 Introduction

Formal verification is essential for the safe deployment of autonomous robots (Tellex et al., 2020; Raman et al., 2013), cyber-physical controllers (Konur, 2013), and safety-critical software systems (Alur, 2015). Verification first begins with a specification that defines intent in precise temporal logic (TL) (Watson & Scheidt, 2005; Bellini et al., 2000). However, human stakeholders typically articulate intent in ambiguous natural language (NL) (Veizaga et al., 2021; Lamar, 2009; Lafi et al., 2021), and the conversion of this NL to TL is a challenging and time-consuming process that requires human experts (Yin et al., 2024; Cardoso et al., 2021; Thistle & Wonham, 1986). Due to this complexity, automated NL-to-TL translation has emerged as a core research problem (Chen et al., 2023; Zrelli et al., 2024; He et al., 2022; Wang et al., 2025). Recently, neural sequence-to-sequence models (Hahn et al., 2022; Pan et al., 2023; Hsiung et al., 2022), grammar-constrained decoders (Post & Vilar, 2018; Geng et al., 2024), and large language models (LLMs) (Xu et al., 2024; Chen et al., 2023; Fuggitti & Chakraborti, 2023; Cosler et al., 2023) have all demonstrated promising results on benchmark corpora, with reported accuracies often exceeding $90\%$.

Despite these gains, evaluations are misleading as most datasets only test *lifted* translation, where temporal logic formulas contain abstract placeholders for atomic propositions (APs). The harder task of *grounded* translation—instantiating APs with domain-specific actions and arguments—is usually left unmeasured. This imbalance stems from limitations of current datasets, which omit the annotations required to separately evaluate lifting, translation, and grounding. As a result, current frameworks optimize for partial tasks, leaving open the more difficult but necessary problem of grounding for producing fully executable specifications.

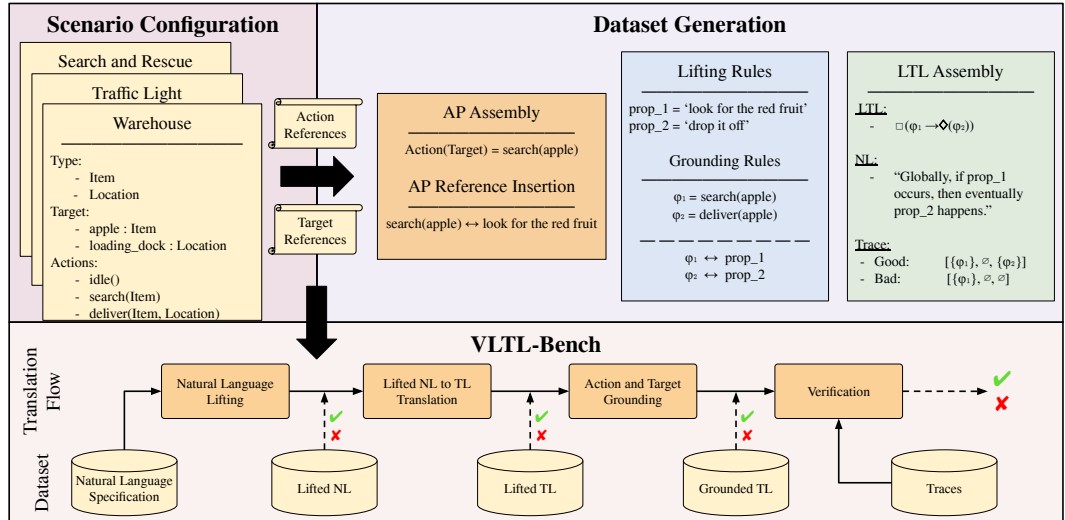

Figure 1: Overview of our dataset synthesis and evaluation framework for NL-to-LTL translation. The framework used a configuration file to define concrete and unique scenarios. The data synthesis generates the NL and TL pairs with associated traces for verification while providing ground truth results for intermediate components.

Benchmarks for NL-to-TL translation include CW (MacGlashan et al., 2015), GLTL (Gopalan et al., 2018), Navi (Wang et al., 2021), and Conformal (Wang et al., 2025). Their limitations are fourfold. (i) Although recent frameworks decompose the task into lifting, translation, and grounding, these benchmarks supply ground truth only for the end-to-end result (NL-TL pairs), preventing assessment of intermediate components. (ii) CW and GLTL omit grounding entirely, yielding translations without executable semantics. For example, the NL specification: "Go to the green room and then go to the blue room." is mapped to the LTL expression: "$\diamondsuit G \rightarrow \diamondsuit B$", without providing a grounded definition of the predicates $G$ and $B$. (iii) Navi and Conformal nominally support grounding but rely on overly simplistic state spaces (e.g., Navi's colored-room grid), which fails to capture the referential and contextual ambiguities of natural language. (iv) Execution traces/trajectories for independent semantic verification (e.g., via model checking), are not provided, preventing rigorous evaluation.

In this paper, we introduce the **Verifiable Linear Temporal Logic Benchmark (VLTL-Bench)**, a benchmark that grounds linear temporal logic (LTL) in a concrete world state space while broadening linguistic and logical coverage through more diverse atomic propositions. As illustrated in Figure 1, VLTL-Bench exposes every stage of the NL-to-TL pipeline: raw and lifted NL specifications, an AP-to-Reference dictionary, lifted and grounded LTL formulas, and both satisfying and unsatisfying traces. Our dataset synthesis and evaluation framework for NL-to-LTL translation leverages scenario configurations to construct grounded action/target combinations, from which we synthesize diverse natural language representations and integrate them into sentence, LTL, and trace templates, yielding corpora whose components can be used individually or combined for holistic evaluation. This layered design makes it possible to isolate performance on lifting, translation, grounding, and verification individually, while also enabling end-to-end evaluation. We provide four scenario configuration files and construct a Kitchen Assistant, Traffic light, Search & Rescue, and Warehouse dataset. Using these four datasets we evaluate the capabilities and limitations of state-of-the-art NL to TL translation frameworks. In summary, we propose: (i) a single, extensible benchmark for evaluating all NL-to-TL translation components; (ii) the first verification evaluation using satisfying and unsatisfying traces; and (iii) an empirical study that reveals both new failure modes in current methods and the severe accuracy decline when grounding is required.

The remainder of this paper is organized as follows. Section 2 covers preliminaries for LTL and model checking. Section 3 contains a detailed description of the Verifiable Linear Temporal Logic Benchmark datasets, as well as details on how they were synthesized. Section 4 includes an evaluation of current NL-to-TL frameworks on both Verifiable Linear Temporal Logic Benchmark and existing datasets. We conclude our paper in 5. Additional details may be found in the Appendix A

## 2 Background and Related Work

In this section we introduce necessary notation and background information on temporal logic systems including terminology, linear temporal logic symbols, and existing NL-to-TL datasets.

**Linear Temporal Logic.** The syntax of LTL is given by the following grammar:

$$\varphi ::= \pi \mid \neg\varphi \mid \varphi_1 \wedge \varphi_2 \mid \varphi_1 \vee \varphi_2 \mid \varphi_1 \Rightarrow \varphi_2$$
$$\mid \bigcirc\varphi \mid \Diamond\varphi \mid \Box\varphi \mid \varphi_1 \cup \varphi_2$$

We further discuss model checking with linear temporal logic in Appendix A.1 and Appendix A.2.

### 2.1 Preliminaries

In this section, we formally define a number of key terms necessary to describe and evaluate NL-to-TL translation systems. In order to provide a cogent description of these systems, as well as a robust evaluation, we define these terms as follows:

**Scenario:** Referred to in existing work as the "World", "Environment", or "Space". A set $S$ of conditions appearing on a trace.

**Condition:** In model checking, a condition is a uniquely-named Boolean variable $c_i$.

**Atomic Proposition:** $\pi \in \Phi$, where $\Phi$ is the set of propositional variables in an LTL expression. During LTL verification, $\pi_i$ is assigned a value by matching with a condition $c \in S$.

**Lifting:** $\lambda$: NL $\to \Phi$, extracting substrings corresponding to APs from natural language.

**Grounding:** $g(\pi) = c$, replacing an abstract AP in an LTL expression with a condition $c \in S$.

**Translation:** $\tau$: NL $\to$ LTL, converting a natural language string into a formal LTL expression.

**Verification:** Given a trace $\sigma$ or Kripke structure $K$, check whether a grounded LTL expression $g(\varphi)$ holds. For trace-based verification, construct a minimally satisfactory $K$ from $\sigma$.

### 2.2 Existing Benchmark Datasets

In this section, we review existing benchmarks for NL-to-TL translation. We compare these corpora in terms of linguistic and logical complexity, and support for evaluation of different framework modules in Table 1. We measure the complexity using the number of unique words appearing in natural language specifications (#Words), as well as the number of unique temporal logic expressions (#TL). In terms of modules, we report if a dataset has support for evaluation of lifting, grounding, and verification. We also provide examples from existing datasets in Appendix A.5.

In Table 1, we observe that the older datasets **Cleanup World (CW)** (MacGlashan et al., 2015) and **GLTL** (Gopalan et al., 2018) from the pre-LLM era have limited complexity both in terms of unique words and temporal logic expressions. While they support evaluation of translation, the lifting data is not explicitly given, the APs do not vary in their form to any meaningful degree, and they can be

Table 1: Comparison of existing LTL benchmarks and VLTL-Bench. We report the number of unique words across all NL specifications and the number of unique LTL specifications. Additionally, we report support for lifting, grounding, and verification.

| Dataset | # Words | # TL | Lifting | Translation | Grounding | Verification |
|---|---|---|---|---|---|---|
| CW (MacGlashan et al. (2015)) | 184 | 37 | $\sim$ | ✓ | × | × |
| GLTL (Gopalan et al. (2018)) | 183 | 37 | $\sim$ | ✓ | × | × |
| Navi (Wang et al. (2021)) | 131 | 6414 | × | ✓ | $\sim$ | × |
| Conformal (Wang et al. (2025)) | 439 | 212 | $\sim$ | ✓ | $\sim$ | × |
| VLTL-Bench Warehouse | 1028 | 5991 | ✓ | ✓ | ✓ | ✓ |
| VLTL-Bench Traffic Light | 217 | 6196 | ✓ | ✓ | ✓ | ✓ |
| VLTL-Bench Search and Rescue | 245 | 5425 | ✓ | ✓ | ✓ | ✓ |
| VLTL-Bench Kitchen Assistant | 376 | 7385 | ✓ | ✓ | ✓ | ✓ |

lexically identified in both the NL and TL elements of each entry ("green room" $\leftrightarrow G$, "blue room" $\leftrightarrow B$, etc.). The **Navi.** corpus, introduced by (Wang et al., 2021), couples NL commands with LTL formulas in a grid world. As Table 1 shows, Navi exhibits a substantial increase in logical complexity, with 6,414 unique formulas and support for partial grounding. Its 221 unique APs make it a strong test of translation and lexical robustness, though this improvement comes at the cost of well-defined grounding rules: the corpus does not specify formal APs, providing instead POS-tagged natural language representations. As reflected in Table 1, the **Conformal** (Wang et al., 2025) dataset introduces 439 unique words and 212 formulas with explicit grounding, but its scale is modest at 1,000 examples. In contrast, VLTL-Bench provides a testbed suited to holistic evaluation across lifting, translation, grounding, and verification. We provide a more detailed quantitative comparison between these datasets and VLTL-Bench in Section 3.4.

## 3 THE VERIFIABLE LINEAR TEMPORAL LOGIC BENCHMARK

In the following subsections, we first introduce *Grounded Scenario Configuration*, which formalizes the world model by defining types, targets, and actions that ensure well-typed logical atoms. We then describe our *Data Synthesis* pipeline, which instantiates expert-crafted NL–LTL templates with scenario-specific atoms to produce paired sentences, formulas, and traces. Next, we present the *Metrics* used to evaluate each stage of the NL-to-LTL pipeline, and finally, we detail the *Datasets* generated from three scenario definitions, highlighting their unique challenges and properties.

### 3.1 GROUNDED SCENARIO CONFIGURATION

To formalize how natural language specifications map onto executable logical structures, we distinguish three interconnected components: **types**, **targets**, and **actions**. *Types* serve as abstract categories that describe what kinds of objects or entities an action can take as input (e.g., a location, an item, or a threat). *Targets* are the grounded instantiations of these action–type combinations, where abstract slots are filled with concrete constants. *Actions* are verbs that capture the capabilities of the agent; each action comes with a signature that specifies the expected types of its arguments. Together, this hierarchy ensures that linguistic expressions can be systematically mapped into well-typed logical atoms: types constrain argument structure, actions define the permissible predicates, and targets bind them to domain-specific instances. Each dataset is parameterized by a *scenario*—a small, declarative world model that provides:

**Types** $t \in \mathcal{T}$: denotes the sort of parameters accepted by an action (e.g. item or location).

**Targets** $\mathcal{L}$: Specific instances of typed arguments, (e.g. an argument apple of type item, or an argument loading_dock of type location).

**Actions** $\mathcal{A}_{\text{args}}$: verbs the agent may perform, which may have one or more targets, (e.g. idle() has no targets, deliver(apple, loading_dock) takes two—item and location).

### 3.2 DATA SYNTHESIS

To produce our datasets, we began with the 36 expert-crafted lifted NL-LTL pairs of the nl2spec benchmark (Cosler et al., 2023), and we added 7 new ones of our own (provided in Appendix A.4). We then transformed these 43 examples into templates to support diverse NL–LTL synthesis. Finally, for each NL–LTL example, we crafted one pair of traces—one satisfying and one violating.

Each dataset entry includes a tuple of these three artifacts,

$$\{ \underbrace{\text{sentence, lifted sentence}}_{\text{NL (raw \& lifted)}} \}, \quad \underbrace{\varphi_G, \varphi_L}_{\text{LTL (grounded \& lifted)}}, \quad \underbrace{\sigma_{good} \models \varphi_G, \sigma_{bad} \not\models \varphi_G}_{\text{Traces (holds \& $\neg$ holds)}} \},$$

and is algorithmically constructed with the following steps:

1. **Template selection.** Uniformly choose a lifted template. Each template has an arity that determines how many atomic propositions must be instantiated.

2. **Atom sampling.** For each argument slot in the template, draw a unique atomic proposition by randomly selecting actions and arguments from the scenario's $\mathcal{A}_t$ and $\mathcal{L}$. Let $k$ denote

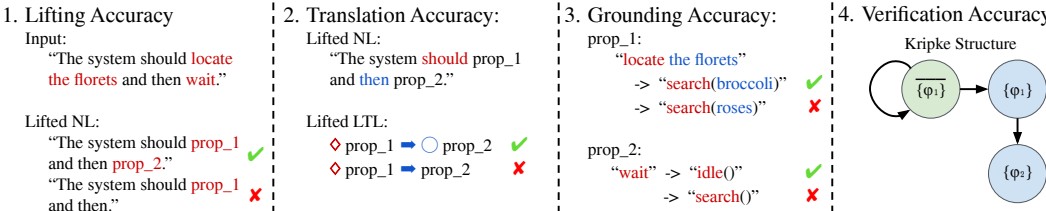

Figure 2: Overview of an isolated evaluation of each individual component. Lifting accuracy measures accuracy of predicted natural language AP spans, grounding accuracy measures the performance on mapping AP spans to world state conditions, translation accuracy measures the performance on NL-LTL translation on the token-level, and verification accuracy is an approach to measuring whether a grounded LTL expression holds on a trace.

    the total number of sampled atoms. Fill the LTL skeleton with these $k$ atoms to obtain the *grounded* formula $\varphi_G$, and replace each atom by $\mathsf{prop}_i$ to obtain the *lifted* formula $\varphi_L$.

3. **NL realization.** Fill the template pattern with each atom's surface form (including articles/prepositions), apply morphological fixes (gerunds, capitalization), and record token-level spans. Emit both the free-form sentence and its `grounded_sentence` with explicit $\mathsf{prop}_i$ placeholders.

4. **Trace filling.** Apply the template's trace patterns to the list $[\mathsf{prop}_1, \ldots, \mathsf{prop}_k]$, yielding one positive trace (satisfies $\varphi_G$) and one negative trace (violates $\varphi_G$).

This rich annotation supports four independent evaluation axes, displayed in Figure 2.

## 3.3 METRICS

In this section, we introduce four complementary metrics that capture performance at different levels of the NL-to-TL pipeline, which is illustrated in Figure 2. Lifting accuracy measures the identification of atomic proposition spans in natural language, grounding accuracy evaluates their mapping to world state conditions, translation accuracy assesses logical equivalence between predicted and reference formulas, and verification accuracy checks whether predicted formulas satisfy or violate traces as expected. Together, these metrics provide a comprehensive view of system performance.

**Lifting accuracy.** For each token $\mathbb{S}_i$ in a sentence, the system predicts a label $\hat{\lambda}(\mathbb{S}_i) \in \{0, 1, \ldots, k\}$, where $0$ denotes background and $n$ denotes membership in $\pi_n$.

$$\text{LiftAcc} = \frac{1}{|\mathbb{S}|} \sum_{i=1}^{|\mathbb{S}|} \big[\hat{\lambda}(\mathbb{S}_i) = \lambda(\mathbb{S}_i)\big].$$

This measures the token-level classification accuracy of mapping substrings to atomic propositions.

**Translation accuracy.** Given a natural language specification $s$, the system produces a predicted TL formula $\hat{\varphi}$. Translation accuracy is an exact match between the predicted and reference formulas:

$$\text{TransAcc} = \big[\hat{\varphi} \equiv \varphi\big],$$

where $\equiv$ denotes logical equivalence. When working with lifted NL, the target is $\varphi_L$; for grounded NL, the target is $\varphi_G$.

**Grounding accuracy.** Let $\{\mathsf{prop}_1, \ldots, \mathsf{prop}_k\}$ be lifted placeholders and $g_{\mathcal{S}}$ the gold grounding function. The system predicts $\hat{g}_{\mathcal{S}}$.

$$\text{GroundAcc} = \frac{1}{k} \sum_{j=1}^{k} \big[\hat{g}_{\mathcal{S}}(\mathsf{prop}_j) = g_{\mathcal{S}}(\mathsf{prop}_j)\big].$$

This measures how well predicted atoms match their reference predicates and arguments.

**Verification accuracy.** For each dataset entry, two traces are provided: a positive trace $\sigma_{good}$ (satisfies $\varphi_G$) and a negative trace $\sigma_{bad}$ (violates $\varphi_G$). Given a predicted grounded formula $\hat{\varphi}_G$, verification

Table 2: Comparison of NL–LTL datasets. We report the total number of entries (Size), the total number of unique TL entries, and the total number of unique APs appearing in the TL entries. †Note that these datasets do not explicitly provide quantities of actions and arguments, and these are estimated by the authors.

| Dataset | Size | Unique TL | # APs | # Actions | # Args |
|---|---|---|---|---|---|
| GLTL Gopalan et al. (2018)† | 11,109 | 37 | 4 | 1 | 4 |
| CW MacGlashan et al. (2015)† | 3,371 | 37 | 4 | 1 | 4 |
| Conformal Wang et al. (2025)† | 1,000 | 212 | 239 | 4 | 235 |
| Navi Wang et al. (2021)† | 7,474 | 6,414 | 221 | – | 26 |
| Kitchen Assistant [VLTL-Bench] | 10,000 | 7,385 | 172 | 9 | 36 |
| Search-and-rescue [VLTL-Bench] | 7,304 | 5,425 | 220 | 7 | 44 |
| Traffic-light [VLTL-Bench] | 7,319 | 6,196 | 5,046 | 4 | 175 |
| Warehouse [VLTL-Bench] | 7,457 | 5,991 | 5,074 | 5 | 82 |

checks whether the satisfaction relation holds:

$$\text{VerifAcc} \; = \; \frac{1}{2}\Big([\sigma_{good} \models \hat{\varphi}_G] \; + \; [\sigma_{bad} \not\models \hat{\varphi}_G]\Big).$$

## 3.4 DATASETS

We construct four scenario definitions accompanied by action and target references, namely a Kitchen Assistant, Traffic Light, Search & Rescue, and Warehouse scenario. The details are provided in Appendix A.8. Using our proposed data synthesis, we generate four new datasets for training and evaluation. Each of our four datasets is designed to highlight distinct challenges for NL-to-LTL translation: the *Traffic Light Control* scenario is intended to balance action and argument grounding challenges, including a large library of "street name" arguments, but a smaller set of actions; the *Search-and-Rescue* scenario emphasizes multi-step temporal dependencies and deliberately includes ambiguous actions such as "avoid" and "communicate" to stress-test the system's ability to distinguish between natural language verbs and temporal operators; the *Kitchen Assistant* scenario includes intentionally ambiguous action and argument references in order to stress both action and argument grounding simultaneously; and the *Warehouse* scenario introduces high semantic and linguistic variability by incorporating all 80 COCO object classes, making grounding especially complex. In this section, we use an entry from the *Warehouse* dataset as an example to illustrate the structure and properties of our data; additional examples from the other scenarios are provided in Appendix A.7.

**Warehouse.** Our *Warehouse* dataset simulates a realistic warehouse retrieval scenario, explicitly designed for scalability and complexity in grounding tasks. Warehouse is our most distinct dataset with its inclusion of all 80 COCO (Lin et al., 2014) object classes, significantly enriching the semantic and linguistic complexity and variation of atomic propositions. As with each of our datasets, all entries include LTL formulas with explicit grounding and alignment at token-level granularity, as well as verified positive ("good") and negative ("bad") execution traces for robust validation.

**Example:**

- **Sentence:** "At every moment, at least one of drop off the long chair to the loading dock, wait, or look for the glass for alcoholic beverage holds."
- **Lifted Sentence:** "At every moment, at least one of `prop_1`, `prop_2` or `prop_3` holds."
- **Grounded LTL Formula:** `globally( deliver(bench, loading_dock) or idle() or search(wine_glass))`
- **APs:** `prop_1` = "drop off long chair to loading dock", `prop_2` = "wait", `prop_3` = "look for glass for alcoholic beverage"
- **Positive Trace:** `[deliver(bench, loading_dock)]`, `[idle()]`, `[search(wine_glass)]`
- **Negative Trace:** `[idle()]`, `[idle()]`, `[search(wine_glass), deliver(bench, loading_dock)]`

## 4 EXPERIMENTAL RESULTS

In this section, we present the results of multiple evaluations of NL-to-LTL translation frameworks and components. In Section 4.1, we measure the performance of common natural language lifting approaches, evaluated on four existing datasets in addition to three of the four datasets we present in VLTL-Bench. In Section 4.2, we evaluate three SOTA NL-to-LTL frameworks on lifted NL to lifted TL translation. Note here, that measuring lifted translation performance on existing datasets is particularly difficult, as they present varying degrees of clarity in their lifted natural language elements. In both translation evaluations, we use the pyModelChecking library (Casagrande, 2024) to determine logical equivalence. The CW (MacGlashan et al., 2015), GLTL (Gopalan et al., 2018), and Navi (Wang et al., 2021) datasets have been processed to include lifted natural language components by (Chen et al., 2023), and we perform similar processing of the Conformal dataset (Wang et al., 2025) to include it in our evaluation. In Section 4.3 we develop and evaluate two grounding baselines on our three datasets. In Section 4.4, we assemble the best results from the three individual evaluations to perform the first end-to-end translation evaluation. In Section 4.5 we perform our novel verification evaluation over the example traces of our dataset.

### 4.1 LIFTING EVALUATION

First, we evaluate four language models on the natural language lifting task. The LLM-based approaches each use the lifting prompt template from the NL2TL framework (Chen et al., 2023), which includes few-shot ground-truth NL to lifted NL examples from each of the datasets. The input to both models is a natural language sentence and we compare the prediction made by the model against the ground-truth lifted natural language using the lifting accuracy metric defined in Section 3.3. We present the results in Table 3 where we see the linguistic complexity of our datasets is highlighted in the accuracies, as even the best scoring model (GPT-4.1) reduces in performance on our new datasets. This performance drop is even more significant on the lower-cost, smaller GPT models. This indicates our success in increasing evaluation complexity.

Table 3: Comparison of lifting approaches.

| | Mean LiftAcc (%) | | | | | | |
| Model | GLTL | CW | CF | Navi | S&R (ours) | TL (ours) | WH (ours) |
|---|---|---|---|---|---|---|---|
| GPT-3.5-turbo | 81.6 | 78.6 | 76.8 | 71.0 | 65.3 | 59.4 | 67.9 |
| GPT-4o-mini | 84.9 | 82.3 | 85.6 | 81.0 | 66.7 | 63.1 | 68.9 |
| GPT-4.1-mini | 97.7 | 95.9 | 96.1 | 97.1 | 94.4 | 96.6 | 93.1 |

### 4.2 LIFTED TRANSLATION EVALUATION

Next, we evaluate the lifted translation capabilities of the three NL-to-LTL frameworks—nl2spec, NL2LTL, and NL2TL. In order to analyze the performance of their lifted translation abilities, the ground-truth lifted NL specification is given to the translation model, and the resulting lifted LTL translation is compared against the ground-truth lifted LTL. The formula for the translation accuracy metric is given in Section 3.3. We present these results in Table 4. Here, we see that lifted translation can be very successful with both out-of-the-box LLM prompting (nl2spec) and with fine-tuned seq2seq models. However, as we have noted, we will see in end-to-end evaluation that this is an overconfident estimation of translation performance as grounding is not considered.

### 4.3 GROUNDING EVALUATION

In this section, we present the results obtained from our evaluation of our baseline grounding framework, applied to the ground truth lifted TL from our three VLTL-Bench datasets. We use two prompting strategies (described in Appendix A.6) applied to three GPT models to provide a broad evaluation of current grounding capabilities. Our first prompting baseline—*few-shot*—is composed of a brief description of the task at hand, accompanied by nine few-shot examples of correct (sentence, lifted sentence, AP-dictionary) tuples from *all three scenarios* (as opposed to individual scenarios).

Table 4: Comparison of four frameworks on the Lifted NL to Lifted TL translation task. Note that we provide ground-truth lifted NL specifications.

| Framework | Model | TransAcc (%) | | | | | | |
| | | GLTL | CW | CF | Navi | S&R (ours) | TL (ours) | WH (ours) |
|---|---|---|---|---|---|---|---|---|
| NL2LTL (Fuggitti & Chakraborti, 2023) | GPT-3.5-turbo | 37.9 | 48.1 | 18.3 | 9.9 | 11.9 | 13.2 | 13.8 |
| | GPT-4o-mini | 38.6 | 55.4 | 23.6 | 10.4 | 12.3 | 13.9 | 12.5 |
| | GPT-4.1-mini | 51.7 | 64.6 | 42.1 | 39.7 | 41.6 | 40.0 | 37.4 |
| nl2spec (Cosler et al., 2023) | GPT-3.5-turbo | 44.4 | 40.9 | 35.2 | 50.3 | 51.1 | 46.3 | 50.2 |
| | GPT-4o-mini | 77.3 | 80.1 | 73.5 | 69.7 | 74.9 | 75.8 | 74.2 |
| | GPT-4.1-mini | 89.8 | 92.9 | 78.3 | 81.5 | 89.1 | 91.6 | 88.4 |
| NL2TL (Chen et al., 2023), Lang2LTL | t5-base | 99.9 | 99.9 | 94.9 | 99.7 | 100.0 | 100.0 | 100.0 |

The next strategy is the *scenario* baseline prompt which includes the full scenario configuration file, as well as three few-shot examples from the dataset. Our final grounding baseline employs the same scenario-specific few-shot examples as the previous approach, with the addition of specific instructions to include intermediate reasoning steps used to arrive at the answer. We use GPT-4o for this chain-of-thought approach in order to evaluate the performance of reasoning-capable models on this task. All four models are instructed to format their final answer in JSON format. To measure grounding accuracy, we parse the resulting AP-dictionary predictions and compare them with our ground-truth knowledge of the AP-dictionary in each entry. Our metrics are per-AP and per-AP-dictionary accuracy. Per-AP accuracy is calculated by recording the total number of correctly grounded APs divided by the total number of APs in the test set, and per-AP-dictionary accuracy is calculated by recording the total number of completely correct AP-dictionaries, divided by the size of the test set. These results are presented in Table 5.

Our evaluation of the two grounding baselines reveals that even advanced LLMs struggle to accurately ground lifted APs into a concrete world state space - even when the parameters of this state space are provided, as is done in the *scenario* baseline. We observe that even though the *scenario* baseline achieves lower performance on most benchmarks and settings, it beats the *few-shot* baseline on our Warehouse scenario when comparing the more powerful reasoning models. As noted in Section 3, the Warehouse scenario is specifically designed to stress-test *grounding and lifting*. We conclude that the provision of the world state space in the *scenario* baseline includes information that aids reasoning models in determining which world state conditions are referred to in the lifted APs, but the overall performance of these baselines on the grounding task remains notably lower than other tasks involved in verifiable NL-to-LTL translation.

Table 5: Comparison of Grounding approaches. This table displays binary accuracy between predicted AP Grounding and known AP dictionary. LLM Baseline uses 9 few–shot sentence + lifted sentence + AP dict examples from every dataset; "Scenario" includes the scenario definition in the prompt and 3 examples from only that dataset. Note that Lang2LTL grounds using cosine similarity between reference and canonical AP embeddings.

| Prompt | Model | Accuracy (% of APs) | | | Accuracy (% of AP Dictionaries) | | |
| | | S&R | Traffic Light | Warehouse | S&R | Traffic Light | Warehouse |
|---|---|---|---|---|---|---|---|
| *Few-shot General* | GPT-3.5-turbo | 56.9 | 69.5 | 18.3 | 34.2 | 51.4 | 7.4 |
| | GPT-4o-mini | 82.3 | 66.5 | 18.4 | 68.6 | 48.4 | 7.0 |
| | GPT-4.1-mini | 77.3 | 67.4 | 23.8 | 60.4 | 45.8 | 7.8 |
| *Few-shot Scenario* | GPT-3.5-turbo | 76.7 | 37.3 | 13.6 | 63.6 | 20.8 | 5.0 |
| | GPT-4o-mini | 66.7 | 44.8 | 23.6 | 44.8 | 16.8 | 9.2 |
| | GPT-4.1-mini | 68.6 | 27.9 | 34.4 | 45.2 | 15.4 | 13.0 |
| *Few-shot Chain-of-Thought* | GPT-4o | 94.8 | 85.9 | 69.9 | 94.1 | 81.5 | 61.4 |
| Lang2LTL (Liu et al., 2023) | *N/A* | 77.6 | 86.2 | 61.8 | 59.0 | 73.6 | 38.8 |

## 4.4 END-TO-END TRANSLATION EVALUATION

Now, we perform and end-to-end evaluation which considers the accumulation of the three individual translation steps. For all three frameworks, we select the best-performing component (model) from

each of the individual evaluations (lifting, grounding, and translation) to assemble an end-to-end translation framework which factors in the combined performance of all the translation steps. We see in Table 6, that as a result of the poor grounding results of all current approaches, the high performance of the lifting and lifted translation steps is diminished, resulting in a poor overall semantic accuracy of the final translation. Our datasets show that even the best performing model (NL2TL) does not approach real-world performance needs, inciting the need for NL-to-TL translation approaches which consider a concrete world state space.

Table 6: End-to-end evaluation of all three SOTA frameworks using the best lifting, translation, and grounding components. We report the binary accuracy of the resulting LTL.

|  | Accuracy (%) | | |
|---|---|---|---|
| Framework | S&R | Traffic Light | Warehouse |
| NL2LTL (Fuggitti & Chakraborti, 2023) | 35.4 | 38.4 | 26.2 |
| nl2spec (Cosler et al., 2023) | 34.8 | 33.6 | 29.6 |
| NL2TL (Chen et al., 2023) | 54.4 | 60.1 | 46.2 |
| Lang2LTL (Liu et al., 2023) | 58.5 | 72.1 | 37.9 |

## 4.5 VERIFICATION EVALUATION

Finally, we present the results of our experiments on the verification of LTL outputs from each of the three NL-to-LTL translation frameworks that we compare. We use the outputs from our lifted translation evaluation (Table 4) to isolate the verification metric from the lifting task, and apply our LLM-baseline grounding frameworks. In Table 7, out results demonstrate that even frameworks exhibiting accurate lifted NL to lifted TL translation suffer a notable decline in performance when grounding relies on systems similar to our LLM baselines. Furthermore, this evaluation supports the use of trace satisfaction in place of ground-truth LTL comparison as a metric for grounded translation accuracy, because the example traces encode the minimum specifications of correctly grounded and translated LTL. In future frameworks, example traces could be used as part of a feedback loop to grounding and translation components.

Table 7: Performance (binary accuracy) on S&R, Traffic Light, and Warehouse, broken down into satisfied holding traces, satisfied not-holding traces, and both. All three frameworks are evaluated on both grounding strategies using their top-scoring lifted translation model.

| Framework | Grounding Strategy | S&R | | | Traffic Light | | | Warehouse | | |
|---|---|---|---|---|---|---|---|---|---|---|
|  |  | Sat | Unsat | Both | Sat | Unsat | Both | Sat | Unsat | Both |
| NL2LTL (Fuggitti & Chakraborti, 2023) | *Few-shot General* | 61.6 | 61.4 | 35.4 | 64.6 | 60.2 | 38.4 | 52.4 | 58.6 | 26.2 |
|  | *Few-shot Scenario* | 1.06 | 32.0 | 7.4 | 61.8 | 59.2 | 36.6 | 12.4 | 36.2 | 9.8 |
| nl2spec (Cosler et al., 2023) | *Few-shot General* | 47.4 | 48.0 | 34.8 | 47.2 | 46.0 | 33.6 | 46.0 | 44.2 | 29.6 |
|  | *Few-shot Scenario* | 34.0 | 36.4 | 21.0 | 40.2 | 41.8 | 28.2 | 32.0 | 34.6 | 19.0 |
| NL2TL (Chen et al., 2023) | *Few-shot General* | 75.0 | 79.4 | 54.4 | 80.2 | 80.6 | 60.8 | 71.4 | 74.8 | 46.2 |
|  | *Few-shot Scenario* | 27.5 | 50.8 | 22.1 | 72.6 | 76.3 | 54.5 | 33.3 | 52.4 | 23.5 |
| Lang2LTL (Liu et al., 2023) | Embedding | 43.3 | 61.9 | 39.3 | 44.7 | 63.0 | 41.3 | 21.6 | 40.1 | 16.6 |

## 5 CONCLUSION

We present the Verifiable Linear Temporal Logic Benchmark. VLTL-Bench is a suite of three new NL-to-LTL translation datasets that include the standard natural language and LTL pairs, supplemented with lifted natural language, lifted LTL, and trace examples. These additional features provide a method for the isolated training and evaluation of individual NL-to-LTL translation framework components. The provision of trace examples in VLTL-Bench introduces the possibility of a new type of input that is plausible in real-world translation frameworks, but unrepresented in current corpora. We acknowledge that the datasets included in the VLTL-Bench suite are generated using a finite number of linguistic and logical templates, populated by diverse synthetic natural language APs. VLTL-Bench reveals significant weaknesses in what were previously ironclad NL-to-LTL translation frameworks. Among these weakness are: the reliance on accurately lifted NL inputs for translation,

lack of accurate grounding components, and lack of example trace inputs in current approaches. We envision our contribution will encourage exploration of diverse methods for grounded NL-to-LTL translation, beyond the use of LLMs.

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

# A  APPENDIX

In this appendix, we present a detailed overview of linear temporal logic in A.1, a discussion of verification via Kripke structures in A.2, a quantitative comparison of our VLTL-Bench dataset against existing datasets as well as examples from those datasets in A.5, our developed prompts for the baseline grounding approaches in A.6, the configuration files for our three scenarios in A.8, and finally our estimated compute resource usage and our external code and license information in A.9.

## A.1  LINEAR TEMPORAL LOGIC

Linear temporal logic (LTL) is a modal extension of classical propositional logic that enables reasoning about how truths evolve over a discrete, linear timeline (Zhu, 2021). Formulas in LTL are interpreted over infinite sequences (or "traces") of states

$$\sigma = s_0, s_1, s_2, \ldots,$$

where each state $s_i$ (which has a set of conditions) specifies which atomic propositions $\pi^\mu$ hold true at time $i$. This framework makes it possible to specify and verify both safety properties (e.g., "nothing bad ever happens") and liveness properties (e.g., "something good eventually happens"), and it underpins many model-checking techniques for reactive systems.

The syntax of LTL is given by the following grammar:

$$\varphi ::= \pi \mid \neg\varphi \mid \varphi_1 \wedge \varphi_2 \mid \varphi_1 \vee \varphi_2 \mid \varphi_1 \Rightarrow \varphi_2$$
$$\mid \bigcirc\varphi \mid \Diamond\varphi \mid \Box\varphi \mid \varphi_1 \cup \varphi_2$$

where $\pi$ ranges over a finite set of atomic propositions; $\neg$, $\wedge$, $\vee$, and $\Rightarrow$ are the standard Boolean connectives; $\bigcirc$ (`next`) asserts that its operand holds in the immediately following state; $\Diamond$ (`eventually`) asserts that its operand holds at some point in the future; $\Box$ (`always`) asserts that its operand holds at every future state; $\varphi_1 \cup \varphi_2$ (`until`) asserts that $\varphi_1$ continuously holds until $\varphi_2$ becomes true. Formally, we write $\sigma, i \models \varphi$ to mean "formula $\varphi$ holds at position $i$ in trace $\sigma$." For example:

$$\sigma, i \models \varphi_1 \cup \varphi_2 \quad \text{iff} \quad \exists k \geq i : \ \sigma, k \models \varphi_2 \ \wedge \ \forall j \in [i, k) : \ \sigma, j \models \varphi_1.$$

Although our focus is on discrete-time LTL, many of these ideas carry over to related formalisms such as signal temporal logic (STL) for continuous-time, real-valued signals (Madsen et al., 2018).

## A.2 VERIFICATION VIA KRIPKE STRUCTURES AND FLUENTS

Verification of LTL specifications is typically conducted using a Kripke structure, which is a formal transition system comprising states, transitions, and labels indicating which atomic propositions hold true in each state. Formally, a Kripke structure is defined as a tuple $M = (S, S\_0, R, L)$, where:

- $S$ is a finite set of states,
- $S_0 \subseteq S$ is the set of initial states,
- $R \subseteq S \times S$ is the transition relation, specifying allowed state transitions,
- $L : S \to 2^{AP}$ is a labeling function mapping states to the sets of atomic propositions that are true in each state.

Verification involves checking whether every possible path through the Kripke structure satisfies the given LTL formula. For instance, safety properties such as *"a collision never occurs"* require that no path through the structure contains a state labeled with the proposition `collision`. Conversely, liveness properties such as *"a goal is eventually reached"* demand the existence of a future state in every valid path labeled with the proposition `goal`. Additionally, verification explicitly involves fluents—timestamped state variables that indicate when certain conditions or states become true. Each fluent captures both the state variable (atomic proposition) and the time step at which the transition into the corresponding state occurs. Formally, a fluent can be represented as a tuple $(\pi^\mu, t)$, indicating that proposition $\pi^\mu$ becomes true at time step $t$ due to a state transition within the Kripke structure. Fluents bridge the gap between high-level temporal specifications and lower-level state transitions, facilitating practical model checking and control synthesis in robot control systems.

## A.3    VLTL-BENCH LTL EXPRESSION STATISTICS

| Token / Operator | Search & Rescue | Traffic Light | Warehouse |
|---|---|---|---|
| and | 5536 | 5508 | 5297 |
| double_implies | 1104 | 1141 | 1091 |
| finally | 3835 | 3842 | 3918 |
| globally | 9899 | 9851 | 9820 |
| implies | 5164 | 5295 | 5264 |
| next | 12144 | 12304 | 11713 |
| not | 6781 | 6724 | 6593 |
| or | 3198 | 3229 | 3054 |
| prop_1 | 14440 | 14466 | 14193 |
| prop_2 | 7934 | 7964 | 7835 |
| prop_3 | 3740 | 3831 | 3825 |
| until | 1112 | 1088 | 1147 |

Table 8: Operator splits and template breakdowns by domain.

## A.4    VLTL-BENCH NEW TEMPLATES

We then craft 7 of our own templates to fill perceived gaps in specification coverage. Of these templates, 4 entries include new lifted LTL halves (marked below with a *), and 3 include new lifted NL halves.

| NL | LTL |
|---|---|
| `finally ( not prop_1)` | "eventually, avoid prop_1" |
| `globally ( not prop_1)` | "always avoid prop_1"; "prop_1 must never occur" |
| `next prop_1` | "at the next time step, prop_1 holds" |
| `prop_1 until prop_2` | "prop_1 must always hold at all times before prop_2" |
| `finally (prop_1 and prop_2)` | OLD: "Eventually, both prop_1 and prop_2 will hold simultaneously"
NEW: "At some point, prop_1 and prop_2 will both hold at the same time." |
| `globally (prop_1 and prop_2)` | OLD: "Both prop_1 and prop_2 hold at every step."
NEW: "At all time steps, prop_1 and prop_2 both hold." |
| `finally (prop_1 or prop_2)` | OLD: "eventually, either prop_1 or prop_2"
NEW: "either prop_1 or prop_2 will hold at some point in time." |

Table 9: Examples of NL–LTL mappings. OLD/NEW entries show updated phrasing.

## A.5    EXISTING DATASETS

**Cleanup World (CW).**

- Sentence: "go to the blue room keep going and stop when you reach the green room"
- LTL Formula: "finally(blue_room and finally green_room)"
- Grounded Sentence: "go to the prop_1 keep going and stop when you reach the green prop_2,"
- APs: prop_1 = go to blue room, prop_2 = go to green room.

**GLTL.**

- Sentence: "enter the blue or red room and proceed until the green room"
- LTL Formula: "finally((red_room or blue_room) and finally green_room)"
- Grounded Sentence: "enter the prop_2 or prop_1 and proceed until the green prop_3,"
- APs: prop_1 = go to red room, prop_2 = go to blue room, prop_3 = go to green room

**Navi.**

- Sentence: "at some time get hold apple or whenever acquire pear"
- LTL Formula: "finally(get_hold_v apple_n or finally(acquire_v pear_n)"
- Grounded Sentence: "at some time prop_1 or whenever prop_2"
- APs: prop_1 = get_hold_v apple_n, prop_2 = acquire_v pear_n

**ConformalNL2LTL.**

- Sentence: "Stay in parking lot 4 until you reach car 5"
- LTL Formula: "parking_lot_4 until car_5"
- Grounded Sentence: "Stay in prop_1 until you reach prop_2"
- APs: prop_1 = go to parking lot 4, prop_2 = go to car 5

A.6  GROUNDING PROMPTS

This section includes the few-shot examples used in our grounding prompt baselines. The *few-shot* baselines uses all of the following in its prompt, while the *scenario* baseline includes only the scenario specific few-shot examples combined with the scenario description, given in Appendix A.8

**Few-shot Prompt:**

"role": "system", "content": "You are an LTL translation assistant, your goal is to return the desired prop_dict, a dictionary that relates natural language atomic proposition/predicate references to their canonical/known representation in the scenario.",
"role": "user", "content":
Few-shot Examples:
`{examples from ALL domains, shown in appendix A.7, total of 9 examples}`
Now predict:
Sentence: `{sentence}`
Lifted: `{lifted_sentence}`
Prop_dict:

**Scenario Prompt:**

"role": "system", "content": "You are an LTL translation assistant, your goal is to return the desired prop_dict, a dictionary that relates natural language atomic proposition/predicate references to their canonical/known representation in the scenario.",
"role": "user", "content":
Scenario Configuration: `scenario yaml, given in appendix A.8`
Few-shot Examples:
`{examples from this specific scenario, shown in Appendix A.7}`
Now predict:
Sentence: `{sentence}`
Lifted: `{lifted_sentence}`
Prop_dict:

## A.7 FEW-SHOT EXAMPLES BY SCENARIO

**Warehouse Examples**

```
Sentence: ["The system must eventually, avoid prop_1"]
Lifted Sentence: ["The system must eventually, avoid prop_1"]
prop_dict: {
"prop_1": {
"action_canon": "deliver",
"action_ref": "drop off",
"args_canon": ["sandwich loading_dock"],
"args_ref": ["square food loading dock"]
}
}
Sentence: ["Whenever prop_1 holds, prop_2 holds as well."]
Lifted Sentence: ["Whenever prop_1 holds, prop_2 holds as well."]
prop_dict: {
"prop_1": {
"action_canon": "idle",
"action_ref": "remain still",
"args_canon": [],
"args_ref": []
},
"prop_2": { "action_canon": "get_help",
"action_ref": "call for help",
"args_canon": [],
"args_ref": []
}
}

Sentence: ["If prop_2 holds, then in the next step prop_3 persists until prop_1 holds, or else prop_3
holds forever."]
Lifted Sentence: ["If prop_2 holds, then in the next step prop_3 persists until prop_1 holds, or else
prop_3 holds forever."]
prop_dict: {
"prop_1": {
"action_canon": "pickup",
"action_ref": "grab",
"args_canon": ["hot_dog"],
"args_ref": ["bunned sausage"]
},
"prop_2": {
"action_canon": "pickup",
"action_ref": "grab",
"args_canon": ["potted_plant"],
"args_ref": ["plant"]
},
"prop_3": { "action_canon": "search",
"action_ref": "search for",
"args_canon": ["cup"],
"args_ref": ["beverage cup"]
}
}
```

**Search and Rescue Examples**

```
Sentence: ["This controller must always avoid prop_1"]
Lifted Sentence: ["This controller must always avoid prop_1"]
prop_dict: {
"prop_1": {
"action_canon": "record",
"action_ref": "begin recording",
"args_canon": ["fire_source"],
"args_ref": ["fire source"]
}
}

Sentence: ["In this task, take a photo of flood, then return home."]
Lifted Sentence: ["In this task, prop_1 then prop_2"]
prop_dict: {
"prop_1": {
"action_canon": "photo",
"action_ref": "take a photo of",
"args_canon": ["flood"],
"args_ref": ["flood"]
},
"prop_2": {
"action_canon": "go_home",
"action_ref": "return home",
"args_canon": [],
"args_ref": []
}
}

Sentence: ["If every record flood is eventually followed by talking to the safe victim, then avoid the
impending debris must occur infinitely often."]
Lifted Sentence: ["If every prop_1 is eventually followed by prop_2 then prop_3 must occur
infinitely often."]
prop_dict: {
"prop_1": {
"action_canon": "record",
"action_ref": "record",
"args_canon": ["flood"],
"args_ref": ["flood"]
},
"prop_2": {
"action_canon": "communicate",
"action_ref": "talk to",
"args_canon": ["safe_victim"],
"args_ref": ["safe victim"]
},
"prop_3": {
"action_canon": "avoid",
"action_ref": "avoid",
"args_canon": ["impending_debris"],
"args_ref": ["impending debris"]
}
}
```

**Traffic Light Examples**

Sentence: ["You", "must", "eventually,", "avoid", "set", "east", "light", "yellow."]
Grounded: ["You", "must", "eventually,", "avoid", "prop_1"]
prop_dict: {
"prop_1": {
"action_canon": "change", "action_ref": "set", "args_canon": ["light_east", "yellow"], "args_ref": ["east light", "yellow"] } }
Sentence: ["Both", "change", "west", "light", "red", "and", "take", "a", "video", "of", "the", "car", "on", "southwest", "10th", "avenue", "hold", "at", "every", "step."]
Grounded: ["Both", "prop_1", "and", "prop_2", "hold", "at", "every", "step."]
prop_dict: {
"prop_1": {
"action_canon": "change",
"action_ref": "change",
"args_canon": ["light_west", "red"],
"args_ref": ["west light", "red"]
},
"prop_2": {
"action_canon": "record",
"action_ref": "take a video of",
"args_canon": ["car", "southwest_10th_avenue"],
"args_ref": ["car", "southwest 10th avenue"]
} }
Sentence: ["If", "take", "a", "picture", "of", "the", "car", "on", "northwest", "6th", "street", "holds", "and", "set", "east", "light", "green", "holds", "next,", "then", "request", "assistance", "holds", "in", "the", "step", "after", "that."]
Grounded: ["If", "prop_1", "holds", "and", "prop_2", "holds", "next,", "then", "prop_3", "holds", "in", "the", "step", "after", "that."]
prop_dict: {
"prop_1": {
"action_canon": "photo",
"action_ref": "take a picture of",
"args_canon": ["car", "northwest_6th_street"],
"args_ref": ["car", "northwest 6th street"]
},
"prop_2": {
"action_canon": "change",
"action_ref": "set",
"args_canon": ["light_east", "green"],
"args_ref": ["east light", "green"]
},
"prop_3": {
"action_canon": "get_help",
"action_ref": "request assistance",
"args_canon": [],
"args_ref": []
}
}

## A.8 SCENARIO CONFIGURATIONS

In this section, we provide the scenario configuration files that are inserted into the grounding prompts and used for data generation.

```
warehouse:

  actions:

    idle:
      role:   ego
      params: []              # idle()

    get_help:
      role:   ego
      params: []              # get_help()

    # one-argument
    search:
      role:   ego
      params: [item]          # search(item)

    pickup:
      role:   ego
      params: [item]          # pickup(item)

    # two-argument
    deliver:
      role:   ego
      params: [item, location]    # deliver(item, location)

  targets:
    item:
      properties: [name]
    location:
      properties: [name]
```

Figure 3: Warehouse Scenario Configuration file

```
traffic_light:

  actions:
   # ego-only
   get_help:
     role:   ego
     params: []                    # get_help()

   # one-argument
   change:
     role:   ego
     params: [light, color]         # change(light_id, color)

   record:
     role:   ego
     params: [target]              # record(target)

   photo:
     role:   ego
     params: [target]              # photo(target)

  targets:
   light:
     properties: [position, color]
   pedestrian:
     properties: [position, status]
   car:
     properties: [lane, speed]
   location:
     properties:  [lane]
```

Figure 4: Traffic Light Scenario Configuration file

```
search_and_rescue:

  actions:
    # ego-only
    go_home:
      role:  ego
      params: []              # go_home()

    get_help:
      role:  ego
      params: []              # get_help()

    # person-centred actions
    communicate:
      role:  ego
      params: [person]        # communicate(person)

    deliver_aid:
      role:  ego
      params: [person]        # deliver_aid(person)

    record:
      role:  ego
      params: [target]

    photo:
      role:  ego
      params: [target]

    avoid:
      role:  ego
      params: [target]

  targets:
    person:
      properties: [injured, trapped, safe]
    threat:
      properties: [active, neutralized]
    location:
      properties: [name]
```

Figure 5: Search and Rescue Scenario Configuration file

```
kitchen_assistant:
  locations: [prep_area, stove, pantry, refrigerator, dining_table]

  actions:
    get_help:
      role:   ego
      params: []                  # get_help()

    wash_gripper:
      role:   ego
      params: []                  # wash_hands()

    retrieve:
      role:   ego
      params: [ingredient, location]   # retrieve(ingredient, location)

    chop:
      role:   ego
      params: [ingredient]           # chop(ingredient)

    cook:
      role:   ego
      params: [ingredient, appliance]   # cook(ingredient, appliance)

    preheat:
      role:   ego
      params: [appliance, temperature]  # preheat(appliance, temperature)

    turn_off:
      role:   ego
      params: [appliance]           # turn_off(appliance)

    turn_on:
      role:   ego
      params: [appliance]           # turn_on(appliance)

    serve:
      role:   ego
      params: [dish, person]          # serve(dish, person)
  targets:
    ingredient:
      properties: [fresh, chopped, cooked, raw]

    appliance:
      properties: [on, temperature]

    dish:
      properties: [ready]

    person:
      properties: [hungry, waiting]
```

Figure 6: Kitchen Assistant Scenario Configuration file

### A.9 COMPUTE RESOURCES AND EXTERNAL CODE AND LICENSE INFORMATION

All LLM inference was performed using the OpenAI API. Approximately \$30.00 in compute credits were used for our evaluations. The T5-base model used by NL2TL was trained and tested locally on a machine using an Nvidia GeForce RTX 4070Ti Super 16 GB GPU, an Intel i9 14900KF, and 64 GB of RAM. Training took approximately 40 minutes using a batch size of 16 and a learning rate of $2e^{-5}$ for 3 epochs.

The nl2spec framework is released at `https://github.com/realChrisHahn2/nl2spec` under the MIT license, the NL2TL framework is released at `https://github.com/yongchao98/NL2TL?tab=readme-ov-file` with no attached license, the NL2LTL framework is released at `https://github.com/IBM/nl2ltl` under the MIT license, and the pyModelChecking library is released at `https://github.com/albertocasagrande/pyModelChecking` under the GNU General Public License.

### A.10 LARGE LANGUAGE MODEL DISCLOSURE

During the preparation of this paper, the authors employed large language models (LLMs) as assistive tools for limited tasks including proof-reading, text summarization, and the discovery of related work. All substantive research contributions, analyses, and claims presented in this paper were conceived, developed, and verified by the authors. The authors maintain full ownership and responsibility for the content of the paper, including its technical correctness, originality, and scholarly contributions.

