# OpenReview forum: "Verifiable Natural Language to Linear Temporal Logic Translation: A Benchmark Dataset and Evaluation Suite"
_ICLR.cc/2026/Conference — Submitted to ICLR 2026_

### Official Review · Reviewer_HCJe · 2025-10-29

**Soundness:** 2
**Presentation:** 3
**Contribution:** 2
**Rating:** 4
**Confidence:** 3

**Summary:**

The main argument of this paper is that while current NL-to-TL systems excel at translating abstract formulas, they fail badly at the crucial step of connecting those formulas to real-world environments (so called "grounding"). To address this, the authors  introduce VLTL-Bench, a new benchmark designed specifically to address this gap and enable end-to-end evaluation of NL->TL translation including verification. This benchmark includes multiple distinct state spaces (environments), Thousands of diverse NL specifications with corresponding formal TL specs,  sample traces for verifying the grounded formulas within each state space and it supports the evaluation of the full process: "Lifting", "Grounding", "Translation", and "Verification"
The authors show that existing systems perform very well on "lifting" and "translation", confirming prior benchmark results, but they struggle significantly with the critical step of "grounding" atomic propositions into a provided state space.

**Strengths:**

-- The paper addresses the major limitation in existing NL-to-TL research: the lack of evaluation for "grounding". Previous benchmarks ignored this essential step needed for real-world verification.

-- The work enables end-to-end evaluation, providing not just formulas, but "sample traces" that allow researchers to verify if the final grounded TL formula behaves correctly in a specific state space/scenario.

-- The paper considers NL-to-TL process as a pipeline of different stages (lifting, grounding, translation) and for each stage it provides the ground truth data.

-- The paper provides a rich & diverse dataset, with "thousands of diverse NL specifications" providing corresponding formal TL specs for each NL spec.

--The paper exposes the specific weakness of existing SOTA models: high accuracy on abstract translation but poor performance on grounding.

-- Broad Evaluation Scope (`nl2spec`, `NL2TL`, `NL2LTL`, `Lang2LTL`) including sequence-to-sequence models and Various LLM prompting techniques.

**Weaknesses:**

-- Scope Limited to LTL: The benchmark specifically targets Linear Temporal Logic (LTL), as indicated by "VLTL-Bench" and references to LTL. This does not evaluate translation  capabilities for other important temporal logics like Computation Tree Logic (CTL) or Signal Temporal Logic (STL).

-- Limited State Space Diversity & Scale:While it includes three distinct state spaces, this is still a small number compared to the vast diversity of real-world systems NL-to-TL might  be applied to. Generalizability beyond these state spaces is not proven.

-- Sample Trace Limitations: While including traces for verification is crucial, their representativeness are very important. The paper does not provide details how these traces were generated or if they comprehensively cover potential system behaviors (e.g., are corner cases included?)

**Questions:**

The benchmark includes only three distinct state spaces. How do you justify that this small number sufficiently captures the complexity and variability  required to evaluate grounding generalization in real-world scenarios? What steps were taken to ensure these environments are not biased or overlapping?

How were the sample traces for verification generated? Can you demonstrate statistically that they cover corner cases, violations, and  satisfactions of the LTL formulas? How do you address concerns about trace completeness impacting verification reliability?

VLTL-Bench focuses exclusively on Linear Temporal Logic (LTL). Many practical applications require richer logics like CTL*, Signal Temporal Logic (STL), or Metric Temporal Logic (MTL). Does your approach fundamentally limit the benchmark's applicability? Is extending it to other TLs feasible?

You attribute poor grounding performance primarily to a 'lack of existing datasets'. However, could the failure also stem from fundamental architectural limitations in current NL-to-TL systems (e.g., LLMs lacking state-space reasoning)? How did you disentangle data scarcity from model capability?

Given the identified grounding bottleneck, what concrete architectural changes, training paradigms (e.g., using VLTL-Bench for fine-tuning), or hybrid approaches do you propose as most promising to address this?

---

> ### Author Response · Authors · 2025-11-21
> **Response to HCJe**
>
> We thank the reviewer for their detailed comments. We would like to address a number of points raised by the reviewer in hopes of clarifying our contributions.
> Firstly, while we recognize the existing need for inclusion of other TL formalisms in the population of NL-TL datasets, our contribution does not attempt to include a broad range of formalisms. We focus our efforts on expanding our ability to train and test a specific type of NL-TL translation system (those which translate into LTL), by including data which is necessary to evaluate individual framework components in isolation. It is crucial to note that no other dataset has done this for any formalism, including LTL, and so we retain our novelty claims on this basis.
>
> >Q1: How do we justify the small number of state spaces?
>
> Please see our answer to Q3 in our response to all reviewers.
>
> >Q2: How are sample traces generated?
>
> Please see our answer to Q5 in our response to all reviewers.
>
> >Q3: Does the approach fundamentally limit benchmark applicability, or is extending it to other TLs feasible?
>
> Please see our answer to Q1 in our response to all reviewers.
>
> >Q4: You attribute poor grounding performance to “lack of existing datasets”, however the failure could stem from fundamental architectural limitations in current NL-to-TL systems. How do you disentangle data scarcity from model capability?
>
> This is a great question, which has a somewhat nuanced answer. The reviewer is correct in that poor grounding performance is due to fundamental architectural limitations of current NL-LTL translation systems. However, these limitations have been unnoticed due to the inability of current benchmarks to perform an isolated evaluation of the grounding capabilities of such systems. We hope that our dataset, benchmark, and evaluation of current frameworks will shed some light on the current fundamental limitations of such systems.
>
> >Q5: Given the grounding bottleneck, what concrete architectural changes/training paradigms/approaches do you propose as most promising to address?
>
> We believe that the addition of a “grounding component” to current framework architectures is necessary to overcome the grounding bottleneck. The majority of current NL-LTL translation systems explicitly assume access to accurate predicate groundings at inference time. Our proposal will allow future works to abandon this assumption and explore avenues for providing these groundings without access to ground-truth information.
>
> We thank the reviewer for highlighting the importance of full-pipeline evaluation. We have added stronger grounding experiments in response to the review, and our results further confirm the necessity of datasets like VLTL-Bench to guide architectural innovations in grounded NL-to-LTL translation.

---

### Official Review · Reviewer_LzLP · 2025-10-30

**Soundness:** 3
**Presentation:** 3
**Contribution:** 3
**Rating:** 6
**Confidence:** 4

**Summary:**

This paper studies an interesting and important problem in the area of natural language to formal language (here is temporal logic). Most previous studies focus on lifted translation, while neglecting the action grounding process. This work finds most models actually fails in Atomic proposition grounds. Hence, they create a new dataset and test the performance of all the SoTA methods. Overall, I think this paper is a good contribution.

**Strengths:**

The great detection of the flaws in current approaches and the solid dataset creation and benchmark testing. The writing and related work articulation are clear. The motivation is great.

**Weaknesses:**

The main question I am doubting is the semantic and expression diversity of the created dataset. The initial 43 expressions seem too limited. Meanwhile, I remember in the referenced work NL2TL, they utilized LLM to help synthesize the initial pairs to then do human annotation. That increases the diversity. In this study, it seems the authors do not utilize LLM for synthesizing. I wonder if the authors can explain what's the reason and possible benefits.

**Questions:**

As I said above, the dataset semantic diversity remains one question.

---

> ### Author Response · Authors · 2025-11-21
> **Response to LzLP**
>
> We thank the reviewer for the positive assessment of our contribution, particularly regarding motivation, identification of grounding as a bottleneck, and the clarity of writing. We respond to the reviewer’s concerns below.
>
> > Q1: semantic limitation and only 43 templates
>
> Please see our answer to Q2 in our response to all reviewers.

---

### Official Review · Reviewer_yG7A · 2025-11-01

**Soundness:** 4
**Presentation:** 4
**Contribution:** 3
**Rating:** 6
**Confidence:** 2

**Summary:**

The paper introduces VLTL-Bench, a benchmark that evaluates natural-language-to-LTL systems end-to-end. The proposed benchmark covers lifting, translation, grounding, and verification across warehouse, traffic-light, and search-and-rescue scenarios. It shows that while models handle lifted translation well, they struggle to ground atomic propositions in concrete state spaces, leading to large drops in end-to-end and trace-verification accuracy.

**Strengths:**

1. VLTL-Bench measures all four aspects and supplies ground truth for each, plus example traces to check whether formulas actually hold.
2. The scenario configs and templated generation make the benchmark easy to extend.

**Weaknesses:**

1. All LLMs used for evaluation all extremely small & non-reasoning LLMs. The task is considered as a reasoning task, so including results for reasoning LLMs will make the evaluation more comprehensive.
2. The benchmark centers on discrete-time LTL. Related logics are discussed in L680 but not supported, limiting applicability to systems needing other temporal formalisms.

**Questions:**

1. I understand that you have provided few-shot examples in A.7, but what's the prompt used for few-shot?
2. Why Actions is limited to a maximum of two targets (as stated in L197)?

---

> ### Author Response · Authors · 2025-11-21
> **Response to yG7A**
>
> We thank the reviewer for their detailed comments. We would like to address a number of points raised by the reviewer in hopes of clarifying our contributions.
>
> As mentioned in our global response, we recognize the general need for datasets and benchmarks over alternative temporal logic formalisms beyond discrete LTL. However, the inclusion of such formalisms is out of scope for this work, as we focus on the robust evaluation of current popular NL-LTL frameworks. Additionally, we include new experimental results using more performant LLMs for our grounding evaluation in the hopes of strengthening our argument for the grounding bottleneck.
>
> > Q1: I understand that you have provided few-shot examples in A.7, but what's the prompt used for few-shot?
>
> We direct the reviewer to Appendix 6 and 7, where all prompts are provided. To make the prompts readable, they were broken into two sections: A.6 and A.7. In A.6 we show the main part of the prompt that wraps the few-shot examples given to the model. A.7 then shows what would be inserted into this prompt for each task.
>
> > Q2: Why Actions are limited to a maximum of two targets (as stated in L197)?
>
> This is not a fundamental limitation, but we consider it a reasonable baseline arity for our scenario predicates. Previous datasets often omit targets (arguments/parameters) from predicates entirely, so we consider the use of arguments in our predicates a significant improvement over current alternatives.
>
> > Q3: All LLMs used for evaluation are extremely small & non-reasoning LLMs. The task is considered as a reasoning task, so including results for reasoning LLMs will make the evaluation more comprehensive.
>
> Please see our answer to Q4 in our response to all reviewers.
>
> Once again, we thank the reviewer for their attention to our proposal and their detailed response, and we hope that we have provided satisfactory clarification where requested, as well as answers to the questions raised.

---

### Official Review · Reviewer_n43H · 2025-11-02

**Soundness:** 2
**Presentation:** 2
**Contribution:** 3
**Rating:** 4
**Confidence:** 3

**Summary:**

This paper introduces VLTL-Bench, a benchmark dataset for evaluating natural language to Linear Temporal Logic (LTL) translation systems. The key contribution is providing ground truth annotations for intermediate translation steps (lifting, grounding, translation, verification) and including verification traces. The authors evaluate several state-of-the-art systems and reveal that grounding (mapping abstract propositions to concrete state spaces) is a significant bottleneck. While the paper addresses an important problem and provides a useful modular evaluation framework, it suffers from limited template diversity (43 templates), insufficient domain coverage (3 scenarios), and, critically, lacks analysis of whether LLMs exploit semantic knowledge versus compositional reasoning. The absence of obfuscated scenario variants (similar to PlanBench [1] for PDDL) is a major weakness that prevents proper isolation of the grounding problem.

[1] Valmeekam et al., PlanBench: An Extensible Benchmark for Evaluating Large Language Models on Planning and Reasoning about Change. NeurIPS 2023.

**Strengths:**

A) The paper correctly identifies a critical gap in existing benchmarks that they focus on lifted translation while ignoring grounding, which is essential for executable specifications. The observation that current datasets achieve >90% accuracy but cannot produce verifiable formulas is valuable.

B) The design allowing isolated assessment of lifting, grounding, translation, and verification is well-motivated and technically sound. The metrics are clearly defined and appropriate.

**Weaknesses:**

A) Limited Template Diversity and Generalization

This is my biggest concern with this work. The benchmark is built from only 43 linguistic and logical templates (36 from nl2spec + 7 new). While thousands of examples are generated through instantiation with different actions/arguments, this approach has some limitations. 43 templates cannot capture the rich diversity of natural language specifications in real-world applications. Human stakeholders express requirements in countless syntactic structures, with varying referring expressions, negations, conditionals, and quantifiers. The high translation accuracy (99.9% for NL2TL in Table 4) likely reflects template memorization rather than robust understanding. Additionally, Table 8 shows a significant imbalance. E.g., "next" appears ~10 times more than "until" across domains. This distribution may not reflect real-world specifications and could bias evaluation toward overrepresented operators. Finally, Models trained and evaluated on template-instantiated data may learn to recognize surface patterns rather than develop compositional understanding.

B) Missing Analysis on Semantic Knowledge vs. Compositional Reasoning

The paper does not adequately address whether LLMs leverage pre-training knowledge for grounding, which is crucial for understanding the grounding bottleneck. It's well known that modern LLMs are trained on massive corpora, potentially including robotics documentation, LTL tutorials, warehouse management systems, and traffic control specifications. Additionally, GPT-4 models may have encountered the nl2spec benchmark (36 templates) during pre-training, artificially inflating performance on lifting and translation while grounding remains poor because it requires scenario-specific knowledge. Can they exploit this semantic knowledge for this benchmark? A possible solution would be to follow PlanBench's approach for PDDL planning. SImilar to it, the paper must include obfuscated versions where semantically meaningful names are replaced with arbitrary symbols. E.g., replace search(apple) with act_3(obj_17) then ask queries like "perform procedure Z on target Q" instead of "look for the red fruit". This would isolate compositional reasoning from semantic shortcuts, and test pure grounding ability based solely on scenario configuration. It might also mitigate data contamination concerns (nl2spec templates are publicly available). This is a major gap that significantly weakens the paper's contributions.

C) Weak Grounding Baselines

Given that grounding is identified as the critical bottleneck (Table 5 shows 5.0%-68.6% AP-Dict accuracy), the baseline approaches are surprisingly simplistic. The baselines use a basic few-shot prompting without even the basic additions like chain-of-thought reasoning, structured output constraints like JSON, iterative refinement with verification feedback using provided traces, and self-consistency or ensemble methods. THis might inflate the efficacy of this benchmark.

**Questions:**

1. Why 43 templates? Can you provide empirical or theoretical justification that 43 templates provide adequate coverage of natural language and logical diversity? What would performance look like with 10 templates vs 100 templates?
2. Will you add obfuscated variants (replacing meaningful names with symbols like "act_3(obj_17)") to isolate compositional reasoning from semantic knowledge? This is standard practice in planning benchmarks (PlanBench) and seems essential here.
3. Have you verified that the nl2spec templates and your scenarios don't appear in GPT-4's training data? Can you test on models with verifiable training cutoffs before your dataset creation?
4. Why didn't you explore chain-of-thought prompting, iterative refinement with trace verification, or fine-tuned models for grounding given it's the identified bottleneck?
5. How were the positive and negative traces chosen for each specification? Are they minimal, typical, or adversarial examples?
6. Current scenarios have flat type systems. Can your framework handle hierarchical types (e.g., "animal -> dog -> poodle")?

---

> ### Author Response · Authors · 2025-11-21
> **Response to n43H**
>
> We thank the reviewer for their detailed comments. We would like to address a number of points raised by the reviewer in hopes of clarifying our contributions.
>
> > Q1: Why 43 templates? Can you provide empirical or theoretical justification that 43 templates provide adequate coverage of natural language and logical diversity? What would performance look like with 10 templates vs 100 templates?
>
> Please see our answer to Q2 in our response to all reviewers.
>
> > Q2: Will you add obfuscated variants (replacing meaningful names with symbols like "act_3(obj_17)") to isolate compositional reasoning from semantic knowledge? This is standard practice in planning benchmarks (PlanBench) and seems essential here.
>
> Our benchmark is specifically designed to evaluate grounding: mapping natural language spans to canonical predicates and arguments. Full obfuscation of predicate and argument names (e.g., replacing search(water) with act_3(obj_17) and “look for the red fruit” with “perform procedure Z on target Q”) removes exactly the semantic signal that any grounding system must use. In such a setting, there is no principled way for a model to align “procedure Z” with act_3 or “target Q” with obj_17; the task collapses into arbitrary label matching rather than compositional reasoning. For this reason, we do not believe fully obfuscated variants are appropriate for evaluating grounding itself, even though obfuscation is useful in planning benchmarks where grounding is already assumed to be solved. In PlanBench, the obfuscated variants serve to break the link between the PDDL symbols and commonsense word meanings, so they can test whether LLMs are actually reasoning over the structure of the planning model rather than free-riding on lexical/world knowledge. While this is useful for evaluating planning and symbolic reasoning capabilities, that kind of obfuscation actually destroys the very thing that a grounding task is meant to test.
>
> > Q3: Have we verified that the nl2spec templates and VLTL-Bench scenarios do not appear in GPT-4’s training data?
>
> While this is impossible to verify for nl2spec, we have prompted the latest GPT model to provide us with these templates without searching the internet. The model (GPT-4o) is able to guess only 5 out of 36 of the NL-LTL pairs. With regard to our own scenarios, we created all four domains after the knowledge cutoff for all LLMs we evaluate.
>
> > Q4: Why didn't you explore chain-of-thought prompting, iterative refinement with trace verification, or fine-tuned models for grounding given it's the identified bottleneck? Given that grounding is identified as the critical bottleneck (Table 5 shows 5.0%-68.6% AP-Dict accuracy), the baseline approaches are surprisingly simplistic. The baselines use a basic few-shot prompting without even the basic additions like chain-of-thought reasoning, structured output constraints like JSON, iterative refinement with verification feedback using provided traces, and self-consistency or ensemble methods. This might inflate the efficacy of this benchmark.
>
> Please see our answer to Q4 in our response to all reviewers.
>
> > Q5: How are positive and negative traces chosen for each spec?
>
> Please see our answer to Q5 in our response to all reviewers.
>
> > Q6: Current scenarios have “flat type systems”, can we handle hierarchical types?
>
> Both the Search and Rescue and Traffic Light scenarios include a hierarchical type system. Note the enumeration of predicates, types, and arguments for these scenarios in Appendix 8 (pages 21 and 22). The “target” supertype is a parameter of the ‘record’ and ‘photo’ and ‘avoid’ action predicates. The “target” type decomposes into 4 subtypes in the Traffic Light scenario and 3 subtypes in the Search and Rescue scenario.
>
> We thank the reviewer for their critical analysis of our proposal, and we hope that our answers to the questions above are satisfactory, and will happily elaborate on any unclear points if necessary.

---

### Author Response · Authors · 2025-11-21
**Response to all reviewers**

We thank all reviewers for their insights and constructive feedback. We appreciate that the motivation and purpose of our paper is clear. We identify four key components of current NL-to-LTL translation systems: Lifting, Grounding, Translation, and Verification. Crucially, we note the widespread omission of grounding and verification components in current NL-to-LTL systems, as well as the inability of current benchmarks and datasets to independently measure the performance of each. In response, we propose VLTL-Bench to measure the capabilities of each component separately. Below, we list some strengths highlighted by multiple reviewers:
- [n43H, HCJe, LzLP, yG7A] Identifying the major gap in current NL-to-LTL systems: they omit grounding and verification.
- [n43H, HCJe, LzLP, yG7A] VLTL-Bench is the first benchmark that enables isolated evaluation of lifting, grounding, translation, and verification components.
- [LzLP, yG7A, HCJe] Clear motivation and strong writing/presentation quality.
To improve our submission, we strengthened the experimental results with additional grounding evaluations using a stronger reasoning model (GPT-4o) and chain-of-thought prompting.

We now address several concerns raised by multiple reviewers:

>Q1: [HCJe, yG7A] Lack of inclusion of other TL formalisms (CTL, STL):

We focus on LTL because our primary goal is to provide the first benchmark that enables independent evaluation of all NL-to-TL components, especially grounding and verification, which no existing dataset supports even for LTL. While our framework is not inherently limited to LTL, extending it to CTL, STL, etc. would require substantially different dataset design choices, so we leave these richer logics to future work.

>Q2: [n43H, LzLP] Limitations of templated data and of 43 templates for diversity:

We intentionally use 43 hand-crafted templates because they reliably generate coherent NL–LTL–trace tuples while guaranteeing logical correctness, syntactic validity, and full control over grounded predicate–argument structures—properties that algorithmic LTL generation and LLM-based synthesis cannot consistently ensure. Template count is not the primary source of diversity: VLTL-Bench derives combinatorial richness from domain variation, hierarchical types, multi-argument predicates, trace variability, and grounded predicates. Empirically, this diversity is sufficient to expose severe grounding failures across current systems. We also note that prior NL–LTL benchmarks use fewer templates and less linguistic diversity (Table 2), whereas VLTL-Bench has significantly more numerous and varied predicates and arguments. While additional templates expand logical coverage, the central challenge posed by VLTL-Bench is the grounding of atomic propositions. Empirically, we observe that this design already exposes severe grounding failures across all baselines, suggesting that the available combinations of APs and traces leave substantial room for future systems to improve rather than saturating the benchmark.

>Q3: [HCJe, LzLP] Small number of domains/state spaces:

We have extended the initial three domains of VLTL-Bench with an additional Kitchen Assistant domain. The domains serve as a proof of concept for our data-generation protocol and demonstrate that grounding can be evaluated independently of lifting and translation. Our goal is to establish the framework and provide initial domains rather than enumerate all possible environments; the benchmark is designed to make future domain extensions straightforward. Adding a new domain requires specifying a scenario configuration file and defining synonyms for canonical actions and arguments; our data-generation code handles English morphology (tense, gerunds, part of speech) to ensure fluent text. Complexity metrics for Kitchen Assistant are provided in Table 1 and Table 2, and the scenario configuration file is included in Appendix 8.

>Q4: [n43H, yG7A] Grounding evaluations are too simple and do not employ reasoning models or CoT:

Our updated grounding evaluation with GPT-4o and few-shot chain-of-thought addresses this concern (Table 5, Section 4.3). This experiment shows that even powerful LLMs cannot fully solve grounding. GPT-4o performs better than other models, achieving 94% accuracy on Search and Rescue, but its accuracy in Traffic Light and Warehouse is notably lower (82% and 62%, respectively), indicating substantial remaining headroom.

>Q5: [n43H, HCJe] How are positive and negative traces chosen for each spec?

The positive and negative trace examples are hand-made, not generated, as stated on line 204 in Section 3.2. They provide a partial correctness metric without revealing the ground-truth LTL: if a proposed LTL expression is not satisfied by the positive trace, or is satisfied by the negative trace, it is known to be incorrect. We provide these traces to encourage future NL–LTL systems to consider scenarios in which they have access to labeled traces.

---

### Author Response · Authors · 2025-12-02
**Message for AC**

VLTL-Bench addresses a critical and previously overlooked gap in NL-to-LTL research: the inability of existing benchmarks to evaluate grounding and verification in isolation. All four reviewers acknowledged this as a meaningful contribution, with the benchmark being the first to provide ground truths for each pipeline stage (lifting, grounding, translation, verification). In response to reviewer feedback, we strengthened our submission by (1) adding a fourth domain (Kitchen Assistant) to address concerns about state space diversity, (2) including new grounding experiments with GPT-4o and chain-of-thought prompting, which demonstrate that even strong reasoning models struggle with grounding (62–94% accuracy depending on domain), and (3) providing detailed clarifications on template diversity, trace generation, and the intentional scope limitation to LTL. The updated results reinforce our central finding that grounding remains a major bottleneck, validating VLTL-Bench's purpose as a diagnostic tool for guiding future architectural improvements. We believe the revised submission adequately addresses the reviewers' concerns while maintaining the paper's core contributions, and that it's likely that the reviewers would have increased their scores had the discussion period continued.

---

### Meta-Review · Area_Chair_yD2o · 2026-01-07

**Summary:**

The paper proposes a new benchmark for evaluating natural language to Linear Temporal Logic (LTL) translation. Its main contribution is providing ground-truth annotations for intermediate steps—lifting, grounding, translation, and verification—along with sample traces for verification. All reviewers agree that the benchmark enables isolated evaluation of different translation stages and the anlysis identify weaknesses in prior methods, particularly in grounding and verification.

The main concern is the scale and diversity of the benchmark. Reviewers 1, 3, and 4 note that the benchmark is based on only 43 templates and covers 3 domains. Reviewers 1 and 2 also point out that the baselines are relatively simple and do not use reasoning models or chain-of-thought prompting. Reviewer 1 further raises the possibility of data contamination, as LLMs may have seen related corpora and could exploit prior semantic knowledge.

In the rebuttal, the authors add one additional domain and argue that the number of templates exceeds those of prior benchmarks. They also suggest that the low grounding accuracy of current models indicates sufficient diversity. Additional results with GPT-4o using chain-of-thought prompting are provided, showing 94%, 82%, and 62% grounding accuracy on each domain. However, these additions do not fully address the concerns about benchmark diversity and potential data contamination, since GPT-4o with chain-of-thought prompting already achieves high accuracy, while more advanced models (e.g., GPT-5, Claude, Gemini, DeepSeek, Qwen) are not evaluated.

Given the mixed reviews and the fact that the major concerns remain unresolved, I lean toward rejection.

**Reviewer Concerns:**

Concerns about Limited to LTL, the construction of sample traces for verification, and comparisons with chain-of-thought reasoning models are addressed.

The major concerns regarding benchmark diversity and potential data contamination remain unresolved.

**Reviewer Scores:**

Reviewers 1 and 4 are unlikely to raise their scores, as their primary concerns about diversity are not addressed.

---

### Decision · Program_Chairs · 2026-01-26

Reject